# African Refugee Youth’s Experiences of Navigating Different Cultures in Canada: A “Push and Pull” Experience

**DOI:** 10.3390/ijerph18042063

**Published:** 2021-02-20

**Authors:** Roberta L. Woodgate, David Shiyokha Busolo

**Affiliations:** 1Rady Faculty of Health Sciences, College of Nursing, University of Manitoba, Winnipeg, MB R3T 2N2, Canada; 2Faculty of Nursing, University of New Brunswick, Moncton, NB E1C 0L2, Canada; David.Busolo@unb.ca

**Keywords:** youth, refugee, qualitative research, adaptation

## Abstract

Refugee youth face challenges in navigating different cultures in destination countries and require better support. However, we know little about the adaptation experiences of African refugee youth in Canada. Accordingly, this paper presents the adaptation experiences of African refugee youth and makes recommendations for ways to support youth. Twenty-eight youth took part in semi-structured interviews. Using a thematic analysis approach, qualitative data revealed four themes of: (1) ‘*disruption in the family,’* where youth talked about being separated from their parent(s) and the effect on their adaptation; (2) *‘our cultures are different,’* where youth shared differences between African and mainstream Canadian culture; (3) *‘searching for identity: a cultural struggle,’* where youth narrated their struggles in finding identity; and (4) ‘*learning the new culture,’* where youth narrated how they navigate African and Canadian culture. Overall, the youth presented with challenges in adapting to cultures in Canada and highlighted how these struggles were influenced by their migration journey. To promote better settlement and adaptation, youth could benefit from supports and activities that promote cultural awareness with attention to their migration experiences. Service providers could benefit from newcomer-friendly and culturally sensitive training on salient ways of how experiences of multiple cultures affect integration outcomes.

## 1. Introduction

Refugee youth form a considerable proportion of the migrant population and their adaptation to the Canadian society is crucial. In 2016, 1.2 million new migrants had settled in Canada in the preceding five years and refugee youth made up 12.4% (approx. 150,300) of them [1]. Migration and the resultant acculturation can be stressful and result in poor psychological and socio-cultural adaptation [2]. 

Refugee youth arrive in Canada with their own cultural norms and beliefs and often struggle to blend in with the culture in Canada [2,3,4]. The youth experience challenges with education [5], careers [6], legal, and employment system [7]. They must work at finding a balance between the culture from their country of origin and the mainstream culture of Canada. To navigate different cultures, refugees may utilize acculturation processes of integration, assimilation, marginalization, and separation [8]. With integration, certain facets from the heritage and mainstream culture are adopted [2]. Assimilation involves immigrants and refugees seeking distance from their heritage, while placing effort into blending into the mainstream culture. Marginalization involves situations where immigrants and refugees may not want to associate with either the mainstream, or their heritage culture. Separation on the other hand, refers to circumstances where newcomers seek to stay with people from their culture while avoiding the mainstream culture [2]. 

Studies of migrants at their final destination countries report that the type of parental support and practices, language proficiency, cultural distance, and experiences with discrimination can shape youth’s adaptation experiences [2,3,4,9,10,11,12,13,14]. For instance, harsh parenting and a lack of parental support for unaccompanied refugee youth leads to adaptation difficulties such as lack of positive supportive relationships [9,10,12]. A notable cultural distance between youth’s home country and country of resettlement is associated with an increased likelihood of experiencing poverty, discrimination, and poor mental health [3,15,16,17]. However, having a strong foundation in one’s original culture and identity can help to navigate the challenges related to cultural distance [10,16,18]. 

To overcome acculturative stress and struggles, research reveals that refugee youth in general can benefit from reshaping their identities and engaging in meaningful activities, as well as having social support, positive emotions, and friends [19,20,21,22]. Research specific to African refugee youth in final destination countries found that they manage acculturative stress by a number of strategies including relying on religion [23,24], maintaining a sense of collectivity and communal support, relying on their families, heavy alcohol use, making sense of the challenges, cultural orientation and focusing on positive thoughts [16,24,25,26,27]. Other African newcomer youth manage acculturation stress by maintaining their heritage such as eating food from their African countries of origin and speaking dialects from their African countries of origin [28]. In addition, African newcomer youth have been found to adapt by learning about new cultures, changing their habits, suppressing their thoughts and memories of past struggles, and becoming self-reliant even when support is available [27,28,29]. 

In a meta-ethnography by Kennedy and MacNeela [21] that examined the acculturation experiences of youth who immigrated from Asia, Middle East, Africa, and the Americas, most of the reviewed studies involved youth who moved to United States. Findings from this work revealed that youth made sense of acculturation based on their life worlds, pre-immigration experiences, cultural identities, and aspirations as shaped by the life domains of family, school, and peers. Youth experienced changes in the role and relationships within their families. When it came to school environments, youth had mixed experiences of receiving support and felt excluded because of language, religion, and skin complexion. Youth adapted to acculturation by keeping ties with peers from their ethnic communities but also found distance between their groups because of divergent beliefs and lacking financial resources.

Refugee youth usually come with a history of living in refugee camps or within informal settlements that can affect how they adapt to their acculturation in their final destination countries [30]. Influencing their acculturation and adaptation experiences are socioeconomic factors, poor access to education, insecurity, policies, and gender divisions that can lead to discrimination, poor health and settlement challenges. While research exists about refugee youth’s adaptation experiences including the way they navigate different cultures in their final destination countries, it mainly involves quantitative methods and non-African youth with a focus on second-generation migrant or refugee youth [21]. Research that examines the adaptation experiences of newly arrived African refugee youth in Canada is critically needed. Such research is crucial in understanding youth adaptation experiences in their destination country. Gaining an understanding of these youth’s experiences can help to inform and improve adaptation and settlement support programs specific to youth. Accordingly, the purpose of this paper is to present research findings that depict the adaptation experiences of African refugee youth in Canada.

## 2. Materials and Methods

### 2.1. Design

In order to arrive at an understanding of youth’s adaptation experiences, a qualitative research study approach was used [31]. A qualitative approach that was cross-sectional in nature helped to ensure a richness of data by affording youth the opportunity to express their migration experiences in ways that are meaningful to them and present their authentic accounts of navigating and adapting to different cultures. 

### 2.2. Ethical Considerations

Prior to collecting data, we took steps to ensure ethical considerations were maintained. First, we obtained ethical approval to conduct this study from the Education and Nursing Research Ethics Board at the University of Manitoba. Second, we obtained consent from youth who were at least 18 years old while we obtained assent from youth who were 17 years old and younger in addition to consent from their parents.

### 2.3. Data Collection

This study was conducted in Winnipeg (population: over 828,000), Manitoba (population: 1.336 million), in mid-Western Canada [32]. Refugee youth (between ages 15 and 29) who migrated to Canada in the preceding six years were recruited through purposive and snowballing sampling [31]. Purposive sampling was used to allow for selection of information rich participants to take part in the study. Snowballing sampling was used to allow youth to refer other African refugee youth who met the recruitment criteria to take part in the study. Similar to Statistics Canada, we describe youth in our study as people between the ages of 15 and 29 [33]. Youth were recruited using posters, information sessions at immigrant and refugee centres and through word of mouth (participating youth referring their peers to the study). Youth took part in semi-structured interviews which created them the space to share what was important to them as well as helping us to arrive at a deeper understanding of their experiences. The interviews which were between an hour to an hour and a half long, were digitally recorded. The interviews were conducted by three research assistants trained and supervised by the first author. Youth were interviewed in English, French, or Kiswahili. In those situations where youth chose to communicate in French or Kiswahili, a certified translator was present to provide translation and back interpretation. At the end of every interview, the research assistants completed field notes about the interview settings, non-verbal communication, and their reflections. In the interviews, we asked youth questions such as “Can you please tell me about yourself?” “What was life like for you and your family before coming to Canada?” “Could you please tell me what it was like for you and your family to come to Canada?” and “What did it feel like when you first got here?” These questions led to youth talking about their life in their countries of origin, their cultural backgrounds, identity, and comparisons between their backgrounds and Canadian culture. Further probing took place to gather more understanding on their adaptation experiences.

### 2.4. Data analysis 

Interviews and field notes data from youth informed data analysis. Digitally recorded interviews were transcribed verbatim. The interview scripts and field notes were read multiple times to get a sense of the data. Then, using careful line-by-line coding, chucks of sentences on the transcripts were assigned codes. The codes were compared, contrasted, and combined to form categories. Continuing with the process of comparing and contrasting, categories were clustered together to form themes [31].

Several measures were taken to ensure that the study was rigorous. We spent a prolonged period of time with the data and utilized reflexivity whereby our reflections and preconceptions were documented on field notes. During data analysis, we revisited our reflections and compared them with study findings. The researchers came from different backgrounds (Caucasian and African descent with expertise in nursing, political studies, and community health) which provided room for approaching the study phenomena from different perspectives [34,35]. 

## 3. Results

### 3.1. Participant Characteristics

We recruited 28 youth who had recently moved to Canada. The sample was arrived at upon realizing that we were coming across redundant information in our interviews which was an indication of data saturation [36]. There were 9 (32.1%) females and 19 (67.9%) males between ages 13 and 25 (Mean = 19.2, SD = 3.52) years. There were 10 (35.7%) adolescents and 18 (64.3%) young adults. Most of the youth were originally from the Democratic Republic of Congo (DRC) (57.1 %), Burundi (14.3 %), and Somalia (7.1 %). The youth arrived in Canada as refugees with 7 (25 %) of them being unaccompanied, 15 (53.6 %) being accompanied by their families while 6 (21.4 %) were accompanied by one of their parents. By the time of carrying out the study, youth had lived in Canada for an average of 4.87 years. 

### 3.2. Youth’s Cultural Adaptation Experiences Before Coming to Canada 

Experiencing new cultures and undergoing the process of acculturation started when youth left their countries of origin. Youth migrated through different transitional countries in Africa where they lived for between 1 and 13 years. Twenty-five (89.3%) of the youth lived in one transitional country, while three (10.7%) lived in multiple countries prior to migrating to Canada. When asked about where they lived after leaving their countries of origin, youth’s responses reflected a journey through countries and was similar to the following quote: 

“*I left Sudan and went to Ethiopia to start school and then I stayed for a couple of years then they gave us a resettlement paper. We filled it out and ended up coming here.*” (25-year old Sudanese male)

Youth moved through countries because of war or other conflicts. At times, their families stayed in more than one place as they searched for an ideal place to settle leading to a sense of instability. In their discourse, youth used phrases and words such as you have to *“move to a different province,” move to a different country,” “you have to go back,”* and *“we never lived permanently in one place,”* to emphasize the instability that living while on the move created. 

Living in multiple places and countries introduced them to new cultures with youth having to deal with multiple ethnicities. One of the youth that migrated to Cameroon from Burundi mentioned:

“*We went to Cameroon, I saw our house, the new house and I was surprised, I was not used to it. It felt so weird. I saw the people and they were all different because there were people from Burundi and Cameroon. Their cultures were also different.*”(20-year old Burundian male)

Youth often, felt discriminated because of their ethnicities and countries of origin. Because of facing discrimination, youth struggled to belong:

“*Well at first, it was kind of hard because people in Nigeria were insulting us by saying, ‘You Congolese people, you guys cannot stay in your country,’ because there are a lot of Congolese people in Nigeria. I felt like the people there were ignorant, I felt like I had nowhere to go, my life was ruined, and I did not know why I was still living.*” (22-year old Congolese male)

The challenges of living in places other than their home country helped to shape youth’s identity and approach to navigating different cultures. For the most part, youth integrated to the cultures and their identity evolved to the point of feeling more connected with the transition countries rather than their birthplaces. When it came to the time of migrating to Canada, youth expected their experiences to be better. Youth expected to tap into their identity and integration experiences of living in multiple countries and finding ways to adapt to different cultures in Canada. 

The youth’s cultural adaptation experiences in Canada are depicted in the four themes: (1) disruption in the family, (2) our cultures are different, (3) searching for identity: a cultural struggle, and (4) learning the new culture.

#### 3.2.1. Disruption in the Family

On the theme of ‘disruption in the family,’ youth talked about separation from their parent(s) because of their experiences of leaving their countries of birth, and its influence on their adaptation in Canada. Youth’s parents separated because of divorce, or because one or both parents moved to a different country in search of opportunities that were missing in their countries of birth. At other times, the disruption was because of loss of one of the parents. One of the youth who was originally from Burundi narrated that his parents separated because his father needed to find better employment:

“*My dad went back to Burundi to work because in Cameroon it was hard for him to work. In Cameroon, it is not easy to find a good job and stay there. In Cameroon he had a nice job, but they did not pay him well. The payment was always late yet in Burundi he had a better position, so he went back in Burundi to work.*” (20-year old Burundian male)

In the initial times after their families were disrupted, youth struggled to understand the reasons for the separations and whereabouts of their parents. Loss of contact with parents resulted in youth feeling ill fated, lonely, and sad. Two refugee youth mentioned:

“*My parents divorced when I was like four years old, so I never saw my father since I was like seven years old. You know to grow up with that longing for my father is not so lucky.*” (14-year old Somalian male)

“*I ask my mom, ‘Who is my father?’ Sometimes she tells me, ‘Your father was a good man, he usually helped me a lot, he was trustworthy,’ all those. But I ask her, ‘Why did he leave us?’ She tells me, ‘I do not know too.’ It makes me wonder.*” (14-year old Somalian male)

Youth coped with the experience of living without their parents by seeking out for opportunities on their own. One of the youth who was left by his parents when he moved to Canada talked about living far away from home and learning to be self-dependent. He migrated to Canada with his sister, but his parents never joined them. They remained in Sudan. He found it difficult to live away from his parents but felt that the experience shaped his perspectives and approach to life:

“*I keep repeating ‘it is a hard time’ because as a kid I did not get that love that one is supposed to get from their parents. I was just going to school and pretty much raised myself. I had to grow up on my own, never stopping to work for something that I wanted. I never gave up or whined for anything. I never ask anybody for anything, but I believe in hard work.*” (25-year old Sudanese male)

When parents separated, either one parent raised the youth or a relative whom they felt affected their future relationships. For example, one of the youth who was raised by her grandmother had difficulties in relating with her stepsiblings and mother when they reconnected in Canada. The teenager felt misunderstood by her stepsiblings and struggled to understand them as well.

“*When I arrived, I joined my family and it took me a long time to connect with my stepsiblings. I knew that they loved me, but I was not used to them. When I came, I just used to live with my cousin, my sisters, and my brothers. I never had a chance to talk to my stepbrothers, and then when I came everybody was happy to see me. However, we could not understand each other. It was so hard. I was so sad and sometimes I thought that maybe my mother did not like me either.*” (18-year old Congolese male)

The family disruptions that took place before and during youth’s resettlement, made it difficult for youth to settle down and adapt to life in Canada. These youth lacked the social support by their parents. 

#### 3.2.2. Our Cultures Are Different

On the theme of ‘our cultures are different,’ youth talked about differences between cultures from their African countries of origin and Canadian cultures that were evident upon their immigration. Youth were relieved to come to Canada but were astonished to face new challenges. The cultural differences between their birth countries and Canada were great. In their discourse, youth talked about the differences as a ‘*culture shock*’ and went further to describe how families, parenting, and food were perceived differently. Youth used words like *‘here,’ ‘there,’ ‘back home,’ ‘in Africa,’* or *‘in Canada’* to elaborate on the cultural differences in the two places.

In youth’s discourse about how families were different, they felt that in Canadian culture, families are more of the nuclear type, which comprises of a father, mother, and children. Whereas in their countries of birth, families were perceived to go beyond the connotations of nuclear types to include community members and neighbours. Because of the family structure in birth countries, one felt a sense of support, sense of belonging, and believed there was someone watching their back while in Canada, there was a sense of loneliness, isolation, and individualism.

Some people have a different concept of family. For example, in the African culture, family does not just mean you, your parents and your siblings. It is basically the neighbours. You consider them as family such that everybody knew everybody pretty much. (16-year-old Burundian Male)

In expressions of familism, refugee youth’s families expressed a welcoming attitude to guests. During data collection, research assistants were often asked to join families in sharing meals, which was common in the culture of many African countries. One of the research staff reflected the welcoming experience where other people are perceived as family members and invited to share meals. In her field notes, she stated: “I was impressed and humbled by the way the participant and her family welcomed me. After the second interview, they insisted that I share a meal with them. I observed from the food that they did not have much prepared, but they insisted that they share the little they had with me.” Their generosity reminded her of a proverb from her country back home which states, “*However little food we have, we’ll share it even if it is only one locust*” [37]. 

Youth felt there were differences between the cultures around parenting at their countries of birth and Canada. At their countries of birth, youth felt the culture around parenting involved disciplining children using stricter and firmer behaviours such as parents “*shouting orders*” at their children. However, the cultures in Canada encouraged parents to be more lenient and supported children to report to the authorities all forms of strict behaviour including shouting. In addition, youth felt that African parents were very protective and sometimes restricted their movement or roles. On the other hand, Canadian parents were less protective and more willing to let their children develop their own independence.

“*Back home, our parents were very strict on us. They would say, ‘Oh do not go out.’ Those were the standards. But in Canadian culture, parents are very relaxed. You know, their children turn 18 and they tell them ‘We will kick you out of the house, go get yourself an apartment,’ but our parents want to keep you even though you are 21, 22, years old because they fear, they do not trust that their child can live on their own. It is more of the culture thing, like a black people thing.*” (25-year old Nigerian Female)

Sometimes the differences in cultural practices created conflict between youth and their parents. While in their countries birth, youth were expected to follow their parents’ commands without questioning. However, once in Canada, youth had a desire to change how they related to their parents, move on, and adapt to Canadian culture, against their parents’ wishes. When asked about the differences in cultural practices around relating to their parents, youth stated:

Participant:“*We are not in Africa anymore; I know we are expected to keep the African culture but that has to stop*.”

Interviewer:“*You do not think it is a cultural thing, like you know when in the African culture children are meant to be sent to do things*.”

Participant:“*Well I believe there is a limit for everything. There is a limit especially when your parent is able to do what he/she is requesting you to do, because that is just being lazy. We are not in Africa anymore and I have to stopsome of the African cultural practices*.” (19-year old Congolese Male)

Despite the conflict that differences in cultural practices around parenting created, the youth and their parents managed the differences by navigating multiple cultures. Youth and their parents learned from peers about other parenting practices, reflected on their culture and preferences, then adopted practices they were comfortable with. 

Cultural foods were different between youth’s countries of birth and Canada. Upon coming to Canada, youth struggled to get used to their new diets. Some of the regular foods surprised youth because they were not usually part of their diets. Youth found it difficult to come across cultural foods (e.g., halal or foods that were permissible to eat based on their religious or cultural values):

“*I discovered that nothing was halal. What they were selling and used to give us for breakfast was not halal. Back home everything you eat is halal of course. What I did was to ask where I could find halal food. We did not want to feed ourselves with something that was not halal because we are Muslims.*” (20-year-old Congolese Male)

#### 3.2.3. Searching for Identity: A Cultural Struggle 

On the theme of ‘searching for identity,’ youth talked about challenges in finding a balance between their heritage and Canadian culture, and the impact it had on their evolving identity. Youth were undecided on whether to keep the culture of their countries of origin, adopt Canadian culture, or come up with their own culture. The struggles felt like a “*push and pull*” with youth grappling with the desire to please others (e.g., other Canadians) or their families. Youth shared what it meant to struggle with how to navigate two cultures: 

“*How do I adapt myself with the Canadian people? What, how do I behave in a manner that gives pride to all immigrants in Canada and to Canadians and to my family? I have not faced something that may be very rough or very rude to myself or from a Canadian citizen or from other refugees like me. I feel comfortable but I face challenges. I do not succeed, as I desire.*” (24-year old Congolese Male)

Youth who lived in other countries for three years or more or lived in multiple countries expressed more complexity in describing their identity. Their comments about the complexity of their identity were similar to the ones shared here by a 20-year old Congolese male: 

“*We went to Rwanda, Zambia, like a lot of countries. People would always look at me and ask, ‘What are you? Are you a Congolese? Do you speak Swahili?’ I would respond, ‘No I am everything you know’.*” (20-year old Congolese male)

Upon coming to Canada, the youth were on a path to acquiring Canadian citizenship, adding another layer on their identity: 

“*Mostly I ask my mom, I asked her ‘What am I?’ and she told me, ‘You are half Canadian, half Kenyan, half Somalian, because we came from Somalia but I was born in Kenya.’*” (14-year old Somalian male)

Language was an important part of their identity and a source of struggle for the youth. When youth arrived in Canada, they felt a need to keep their traditional language skills from their African countries of origin even though they were no longer in those countries. As a way of maintaining their heritage, youth felt compelled to communicate in their native languages with their friends and relatives from their societies of origin. However, youth found limited opportunities to speak their African dialects with people in Canada considering English or French are the official languages of Canada. 

Another form of struggle with a new culture arose in youth discussions about finding friendships that reflected their identity. Because of migration, youth separated from their friends in their societies of origin and found the need to make new ones in Canada. Almost all youth talked about seeking and keeping friends who shared similar behavior, which was an important aspect of their identity. However, while finding friends was critical, youth struggled with identifying peers that made them feel comfortable. A 16-year-old female from Congo underscored the difficulty she and other youth faced as a young person trying to make new friends in a new sociocultural environment. In her interviews, she kept reiterating *“it is difficult to make friends here. Canadians are different from people in Germany and France*.” Her response highlighted one of the many challenges that refugee youth face when seeking to form new friendships, seeking people they could identify with, as well as find a sense of belonging and acceptance amongst their peers. 

Youth felt indifferent and did not identify with the culture around making friends in Canada. In their perspectives, the culture around making friends with newcomers in Canada was different from what they were used to in their birth countries. In Canada, youth felt they could not approach their neighbours or strangers to create friendships: 

“*You cannot go to someone’s house you know. When I came here, I learned that one needs to give everybody his or her space. You do not go to people. So it was really hard. When you go to school you see everybody is cool you know, they have their friends, like everybody is with somebody, they will not go to someone who is alone. Like nobody really cares about you apart from the teacher.*” (19-year old Congolese female)

Despite the struggle to develop or find their cultural identity, youth found ways of learning and establishing themselves. 

#### 3.2.4. Learning the New Culture

The theme of ‘learning the new culture’ refers to youth experiences of being acquainted with and adapting to different cultures in Canada. Youth needed to find ways to adapt their own culture to the cultural practices at their new country. Youth felt the need to learn the cultures in order to integrate. Youth talked about experiences of learning about the Canadian culture through patience, silence, and careful watching of people’s behaviours. One of the youth expressed his experiences as follows:

“*In my first two months, I was a very quiet kid in school. I was just analysing everything, how people act, because our cultures are very different. I was trying to understand the behaviours of Canadians, how, why, and what they are like.*” (20-year old Burundi male)

Youth adapted to the culture in Canada by tapping into their experiences and understanding of what could help them to navigate different cultures. Youth went to schools where they learned English, found fellow Africans, formed friendships, and learned from them. Youth were better prepared, a little older, and had experiences that could act as reference points for their adaptation and in navigating cultures in Canada: 

“*When we came here, oh my God, I had that feeling for the second time of my life of coming to a new place and living a new life. However, this time, I was older and better prepared. I was like wow, everything looked different, I went to school, and it was better than the school in Cameroon. I really adapted well because usually for newcomers when you go to school and do not know how to speak English, you are not going to make any friends, unless there are some other African that are going to take you and you guys start hanging out. It is sort of like in Africa because in Cameroon, when you are a new student, people come around you, hey what is your name and stuff like that.*” (20-year old Burundian male)

In addition, in Canada, their teachers or immigration counsellors taught youth. Youth were made aware of other cultures, were asked to be respectful, open minded, and willing to learn. Younger siblings were taught about the way of life and how to adapt in Canada by their older siblings. One of the youth shared his experiences of teaching his siblings about Canadian way of life: 

“*I am teaching my brother and sister how to live here. Before they do anything, I ask them to think about their actions and evaluate whether they are good or bad. To ask, ‘Can I do it; if I do it can I be responsible for it?’*” (20-year old Burundi male)

In learning a new culture, youth believed age at immigration and time spent in Canada played an important role. By the time of carrying out our study, youth had lived in Canada for between 1 and 16.5 (average of 4.87) years. Youth felt that those who immigrated at a young age were more likely to learn faster about the Canadian culture with ease compared to older youth. Youth learned to develop their own culture by blending multiple cultures. For example, a youth from Sudan shared his experiences of coming to Canada when he was young and believing that age played an important role. 

“*I grew up here, so I know more but when I came, the experience was just different. English wise, the lifestyles, what I see it is just as if ‘oh I got myself blended in this.’ The good thing I was happy about is that I came here when I was young, and this was great for me you know, I could do a lot of stuff.*” (25-year old Sudanese male)

In addition to their experiences, youth made recommendations for other refugee youth to live in neighbourhoods where other newcomer youth lived. Living in such neighbourhoods could help them to find ways to learn and blend with the Canadian culture at their own pace.

## 4. Discussion

Our study examined the adaptation experiences of first-generation African refugee youth upon their migration to Canada. From youth’s discourse, their experiences focused on how they navigated Canadian and cultures from their African countries of origin. Youth shared their struggles and opportunities they experienced in their integration process. Accordingly, we identified the themes of ‘disruption in the family,’ ‘our cultures are different,’ ‘searching for identity: a cultural struggle,’ and ‘learning the new culture.’ 

Research examining the migration journey and the effect on youth adaptation in receiving countries [3,38,39] is increasing. Much of the work focuses on post-migration challenges that highlight language proficiency, inter-generational conflicts, educational challenges, and difficulties in navigating different cultures and stops short of describing how youth navigate these challenges [38,39]. Our study findings compliment this body of knowledge by highlighting that youth’s migration experiences play a key role in guiding how they navigate different cultures and adapt in their final destination countries. 

Youth in our study described their adaptation experiences in Canada and made suggestions on the need to get used to multiple cultures. Upon migrating to Canada, youth faced difficult experiences of navigating between cultures but learned to live with them. In their integration, youth utilized their experiences of living with new cultures in their migration journey to navigate living with the Canadian culture. While facing different cultures that could be perceived as similar experiences irrespective of where youth migrated to, cultural aspects that youth needed to adjust to be nonetheless different. For instance, youth needed to adjust to the culture around parenting or food when they arrived in Canada. 

On the theme of disruption in the family, youth talked about separation from their parents and the effect it had on their adaptation in Canada. Research examining disruptions in the family because of parental migration reveal experiences of economic challenges, emotional problems, and poor health [40,41,42]. Where family cohesion is present, Filipino adolescents felt supported, had great academic ambitions, and mental health [43]. In our study, family disruptions often took place during times of war with some of the youth never reuniting with their parents. In order to provide support to immigrant and refugee youth with a history of family disruptions, health, and social service providers in receiving countries need to help the youth attain smoother integration experiences. The service providers could connect them with welcoming families from their ethnocultural groups where they could form supportive relationships. 

On the theme of our cultures are different, youth highlighted the culture around families, parenting, and food as areas where cultures from their societies of origin and Canadian differed. From youth’s perspectives, in cultures from their societies of origin, there is a sense of togetherness or familism while in Canadian culture; there is a sense of individualism. When discussing about the culture around parenting, youth felt their parents were strict and authoritarian while Canadian parents were easy going. Arab immigrant adolescents in Ontario, Canada shared similar perspectives and felt that their experiences led to intergenerational conflicts and acculturation stress [44]. However, the Arab immigrant adolescents believed their parents meant well and cared about their future despite the conflicts while youth in our study felt their parents did not want to adapt to Canadian culture or were concerned about their families’ reputation. In future, research that examines African parents’ experiences of parenting in Canada is needed to inform strategies to help parents and their children adapt to Canadian practices. 

Refugee youth in our study had recently migrated to Canada and their perspectives on differences between cultures were in contrast to perceptions by second-generation youth in Toronto [45,46]. The youth in Toronto expressed a sense of belonging that could have been possible because of parental influence and living and attending schools in multicultural neighbourhoods. To navigate the cultural difference, newcomer youth and their families should be encouraged to take part in activities and events that bring youth from multicultural neighbourhoods together. Such activities could include taking part in ethnocultural and national activism organizations [47]. Service providers could also place emphasis in promoting cultural awareness between newcomer and non-newcomer youth to promote better navigation between cultures. 

On the theme of searching for identity, youth faced challenges in negotiating their identity when straddled between two cultures. Youth construct their identity based on their pre-immigration experiences, parents’ identities or culture, personal characteristics, and their environment [46,48,49]. Youth with a history of living in multiple countries struggle and have more complex ways of describing their identity, which can lead to poor mental health [10,49,50,51]. Therefore, when refugee youth arrive in Canada, it is critical to learn about their migration experiences and histories of living in other countries in order to help them adjust to their new way of life that could shape their identity. Youth’s experiences are synonymous with feelings of ‘othering’ which can create tremendous pressure to conform to Canadian culture and identity and poor mental health. Identity struggles can be worsened by government policies that infringe on youth’s culture, religion, or identity (e.g., a ban on the use of hijabs by the Quebec policymakers) [17,52]. To promote youth’s health, migrant settlement organizations and communities needs to create environments that encourage youth to identify with and maintain ties with their native culture. The organizations need to support youth to establish stable social support systems (e.g., friendships or strong family ties) and focus on training programs that support youth’s sense of identity, mental health and wellbeing [53].

On the theme of learning the new culture, youth in our study shared experiences of how they learned and shared knowledge with others. Similarly, unaccompanied minors in Ireland adjusted to life in their new country by learning [28]. While youth in our study often took upon themselves to find ways to integrate, they needed a supportive environment. Similarly, Somali youth in United States expected others (e.g., the government) to help them to integrate [11]. Kennedy and MacNeela [21] found that youth blended with cultures in their resettled countries by learning from peers from their ethno cultural communities. Therefore, to promote youth adaptation, initiatives that promote different ways of youth learning such as learning through social interactions, as well as support from mentors, social service providers, and community members could be beneficial. Helping youth to understand and appreciate living with multiple cultures is warranted. 

Youth in our study emphasize the challenges experienced in navigating between cultures and the impact on their settlement and adaptation in their destination countries. Despite those challenges, researchers elsewhere have found that a strong foundation in one’s original culture and identity can help youth in their adaptation process. Youth can benefit from parental influence and maintaining connections with their culture as they adapt new cultures [46,54]. In the wake of increasing immigration between countries, it is critical to understand and support refugee youth’s adaptation efforts. 

### Strengths and Limitations 

The strength of our study is that we engaged 28 youth from diverse backgrounds to arrive at study findings. We included the languages of English, French, and Kiswahili in our interviews and worked with translators and back interpreters to allow youth to present their perspectives in a language they were most comfortable with. In spite of the strengths, we did not examine refugee youth’s perspectives over time. We were not able to show how youth’s perspective evolved overtime. Also, although youth in our study were of different ages, their experiences were similar. Needed is longitudinal research that examines how refugee youth’s experiences change over time after arriving at their final destination country. Such research could also examine for differences or similarities based on age and sex. 

## 5. Conclusions

African refugee youth arrive in Canada with their own cultural beliefs and experience challenges in adapting to the culture in Canada. These youth present with challenges of family disruptions, cultural conflicts, and their evolving identity, yet they receive limited support. In the wake of increasing immigration because of reasons that include war, refugee youth could greatly benefit from opportunities to develop cultural awareness with attention to experiences of facing new cultures. Additionally, services providers including those who provide social, educational, and health supports could pay greater attention to the unique needs of refugee youth especially those who are relieving adaptation challenges and offer better settlement support.

## Data Availability

Due to ethical restrictions related to protecting participant privacy imposed by the University of Manitoba Education/Nursing Research Ethics Board of the University of Manitoba, the full, qualitative dataset (i.e., interview transcripts and field notes) cannot be made publicly available. Public availability would compromise patient confidentiality or participant privacy.

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
