# Peer review of "African Refugee Youth’s Experiences of Navigating Different Cultures in Canada: A “Push and Pull” Experience"

_ijerph, 2021, doi:10.3390/ijerph18042063_

Round 1

Reviewer 1 Report

It is refreshing to see a paper that explores this unique and important topic.  This manuscript incorporates a thematic analysis approach to describe challenges refugee youth face in navigating different cultures in destination countries such as Canada and the need for better support

The authors assert a need for this study by discussing that refugee youth compose a considerable proportion of the immigrant population in Canada and that this migration and accompanying acculturation can result in poor sociocultural and psychosocial adaptation.

Authors further note that although research exists on refugee youth’s adaptation experiences, the studies are mostly quantitative.  Further, there is lack of studies on non-African youth, particularly second-generation immigrant, or refugee youth. 

The fact that youth were interviewed in English, French, or Kiswahili is important and a strength of the study.

Since interviews were between an hour to an hour and a half long, were incentives provided for participants to participate?  In the case where there were no monetary incentives, were there other considerations, such as providing refreshments or the like to acknowledge participants’ time?

On page 3, it is mentioned that the researchers come from different backgrounds which contributes to the rigor of the study.  The authors should state what the different backgrounds are.

On page 4, the statement - “We went to Cameroon, I saw our house, the new house and I was surprised, I was not used to it. It felt so weird. I saw the people and they were all different because there were people from Burundi and Cameroon. Their cultures were also different. (20-year old Burundian male)” should be in quotations since it seems to be a direct quote.  In other parts of the paper, direct statements are in quotes in some areas and not in quotes in other areas.  Either way, it should be consistent throughout the paper.

Overall, the findings from the participant interviews are compelling and profound.  The conclusions and recommendations include good strategies, such as suggestions for social service providers.

Altogether, this is an interesting, novel, and insightful study.  It is well-written, comprehensive, and well-researched.  There are a few clarifying questions, including about the intervention design, that the authors can explain.  Otherwise, this is a very sound paper.

Reviewer 2 Report

The article discusses African refugee youths’ experiences of navigating different cultures in Canada.  The manuscript also expands upon important knowledge of a vulnerable group of people with respect to race/ethnicity as well as migration status; and it is very timely given Black History Month.  The article is generally written and articulated well.   However, I feel that there are things missing that could strengthen the paper further:

  1. The line numbering has been removed in this version that I reviewed. Please keep the line numbering.
  2. Regarding Paragraph 1: Immigrants and migrants, including refugees, seem to be conflated.  In Canada, immigrants usually refer to economic immigrants who arrive based on skills and experiences.  Migrant, on the other hand, is a broader term that includes both economic immigrants as well as migrant refugees and undocumented migrants.  Perhaps start with a broader term, such as migrant instead of immigrant?   
  3. Page 2, first sentence, re: “At their final destination countries, studies report” This seems awkward and it seems like studies have final destinations. Perhaps revise to “Studies of migrants at their final destination countries report that”. 
  4. Page 2, bottom of paragraph 1 – This part, as well as the discussion section, could benefit immensely from the work of researchers who have contributed to the literature about migrant’s assimilation, accommodation, and integration, as well as the mental and psychosocial health implications when migrants who are utilizing particular cultural practices are forced to shed them, or assimilate to the norms and values of the dominant group.  It may be worthwhile to cite the following work: Syed, I.U. (2013). Forced assimilation is an unhealthy policy intervention: The case of the hijab ban in France and Quebec, Canada. International Journal of Human Rights. 17(3): 428-440. doi: 10.1080/13642987.2012.724678    Also noteworthy could be L. Hessini, ‘Wearing the Hijab in Contemporary Morocco: Choice and Identity’, in Reconstructing Gender in the Middle East: Tradition, Identity, and Power, eds F.M. Gocek and S. Balaghi (1994). New York: Columbia University Press.
  5. Page 2, Paragraph 3, a couple of words are crossed out. Was this by accident? Please revise.
  6. Page 4, re: “We went to Cameroon”…this part needs appropriate quotation marks and any other quotes within quotes require single quotation marks. Please revise.
  7. Page 4, re: “Well at first it was kind of hard” this part needs appropriate quotation marks and any other quotes within quotes require single quotation marks. Please revise.
  8. Page 5, re: “My dad went back to Burundi”, “My parents divorced”, “I ask my mom” “I keep repeating” “When I arrived”…all of these statements need appropriate quotation marks and any other quotes within quotes require single quotation marks. Please revise.
  9. Page 6, re: “Back home, our parents were very strict on us”…this part needs appropriate quotation marks and any other quotes within quotes require single quotation marks. Please revise.
  10. Page 7, re: “I discovered that nothing was halal.” This part needs appropriate quotation marks and any other quotes within quotes require single quotation marks. Please revise.
  11. Page 7, re: “How do I adapt myself with the Canadian people?”, “We went to Rwanda, Zambia” “Mostly I asked my mom”… all of these statements need appropriate quotation marks and any other quotes within quotes require single quotation marks. Please revise.
  12. Page 8, re: “You cannot go to someone’s house you know”, “In my first two months, I was a very quiet kid in school” “when we came here, oh my God, I had that feeling” and “I am teaching my brother and sister how to live here”… all of these statements need appropriate quotation marks and any other quotes within quotes require single quotation marks. Please revise.
  13. Page 9, re: “I grew up here, so I know” This part also needs appropriate quotation marks, see prior comments above.
  14. Re: Discussion section – This is an extremely important section.  I think you should connect some of the findings with implications about what it means for these youths’ identities and their psychosocial/mental health.  Perhaps connect with the work of scholars who discuss the sociocultural status of Aboriginal peoples, immigrants, and refugees, and how there is a process of differentiating, essentialization and significant pressures to conform to Eurocentric values and norms (due to Canada’s colonial/nationalist project).  It may be worthwhile to cite the work of Syed, I. (2020). Hijab, niqab, and religious symbol debates: Consequences for health and human rights. International Journal of Human Rights. Available from: https://doi.org/10.1080/13642987.2020.1826451
  15. Also noteworthy could be: E.F. Isin and M. Siemiatycki, (2002). ‘Making Space for Mosques: Struggles for Urban Citizenship in Diasporic Toronto’, in Race, Space, and the Law: Unmapping a White Settler Society, in S.H. Razack (ed). Toronto: Between the Lines, pp. 185–209.

If you decide to incorporate these revisions, please upload a manuscript that contains tracked changes or other method to highlight revisions.  Thank you for the opportunity to review this work.

Round 2

Reviewer 2 Report

The article discusses African refugee youths’ experiences of navigating different cultures in Canada.  The manuscript also expands upon important knowledge of a vulnerable group of people with respect to race/ethnicity as well as migration status; and it is very timely given Black History Month.  The article is written and articulated well. The authors have followed all of my recommendations for revision and addressed all queries.  The manuscript has improved significantly.  I recommend that it should be accepted for publication

Congratulations, and thank you for the opportunity to review your work.